# PrivAuditor: Benchmarking Privacy Vulnerabilities in LLM Adaptation Techniques

**Derui Zhu**[1*]    **Dingfan Chen**[2*]    **Xiongfei Wu**[3]    **Jiahui Geng**[4]
**Zhuo Li**[3]    **Jens Grossklags**[1]    **Lei Ma**[5,6]
[1]Technical University of Munich    [2]Saarland University
[3]Kyushu University    [4]MBZUAI
[5]The University of Tokyo    [6]University of Alberta

## Abstract

Large Language Models (LLMs) are recognized for their potential to be an important building block toward achieving artificial general intelligence due to their unprecedented capability for solving diverse tasks. Despite these achievements, LLMs often underperform in domain-specific tasks without training on relevant domain data. This phenomenon, which is often attributed to distribution shifts, makes adapting pre-trained LLMs with domain-specific data crucial. However, this adaptation raises significant privacy concerns, especially when the data involved come from sensitive domains. In this work, we extensively investigate the privacy vulnerabilities of adapted (fine-tuned) LLMs and benchmark privacy leakage across a wide range of data modalities, state-of-the-art privacy attack methods, adaptation techniques, and model architectures. We systematically evaluate and pinpoint critical factors related to privacy leakage. With our organized codebase and actionable insights, we aim to provide a standardized auditing tool for practitioners seeking to deploy customized LLM applications with faithful privacy assessments.

## 1   Introduction

The rapid evolution of large language models (LLMs) has made them fundamental to many modern natural language processing tasks [1, 2]. These capabilities are typically powered by vast amounts of model parameters, scaling to trillions, and intensive pre-training on massive text corpora (e.g., nearly a terabyte of English text [3]). However, the large-scale pre-training required for these models incurs significant computational costs, making it financially prohibitive for most practitioners. Additionally, pre-trained models often need additional fine-tuning to achieve satisfactory performance in specific domains [4, 5, 6]. Consequently, the current best practice involves acquiring an open-source LLM as a pre-trained foundation model and then adapting it for domain-specific data.

However, the common "*pre-training, adaptation tuning*" pipeline inadvertently raises privacy concerns regarding the leakage of sensitive domain data used for adapting pre-trained LLMs [7, 8, 9, 10, 11]. Indeed, recent research has demonstrated that LLMs can memorize substantial volumes of sensitive data, leading to a high risk of unintentional privacy leakage to third parties [12, 13, 14]. These issues contribute to the ongoing debate about the privacy implications of LLMs and may trigger violations of modern privacy regulations, e.g., the General Data Protection Regulation (GDPR), underscoring the urgent need to address the privacy challenges associated with LLMs.

To analyze the privacy issues related to the usage of LLMs, existing research primarily focuses on the leakage of pre-training data when querying a deployed general-purpose LLM [12, 14, 13]. Building on this foundation, in-depth investigations regarding such leakage, with respect to various factors

---

*Equal contribution

including model size and the degree of training data repetition, have been presented [10, 15, 16, 17]. Yet, in the context of fine-tuning/adaptation scenarios, recent privacy risk assessments have typically been limited to specific model architectures (mainly encoder-based models), a narrow selection of fine-tuning methods, and a certain choice of attack methods [7, 8, 9, 10, 11, 18]. A comprehensive benchmark evaluation is still missing, despite its importance for providing critical insights and accurate privacy assessments to facilitate the practical application of domain-specific LLMs. In particular, this gap highlights a crucial research question: *To what extent, and in what ways, do different adaptation methods influence the privacy risk of LLMs?*

To address the research question, this paper presents, to the best of our knowledge, the first benchmark investigating the privacy implications of LLM adaptation techniques, accompanied by a comprehensive empirical study. We focus on membership inference attack (MIA) techniques [19], which aim to determine whether a given query sample was used for adapting the target LLM, due to their popularity and close relationship to a broader class of topics [12, 20, 21]. Our investigation encompasses five types of LLMs with different architectures (T5 [3], LLaMA [22], OPT [23], BLOOM [24], and GPT-J [25]), seven LLM adaptation techniques representative of the current state of the art, and three datasets from different domains that closely mimic real-world sensitive fields. With our presented benchmark and comprehensive study, we aim to provide critical insights into the privacy risks associated with LLM adaptation techniques and guide the secure development of new models.

## 2 Privacy Measurement for Large Language Models

We evaluate the privacy vulnerabilities of LLMs through the lens of MIAs [19], which are widely recognized for their extensive applicability. MIAs are also closely associated with other privacy concerns, such as training data reconstruction [12, 15] and the retrieval of personally identifiable information [13, 26, 14], underscoring its critical role in privacy assessments.

### 2.1 Formulation

**Notation.** We denote $f_\theta$ as the target language model, parameterized by $\theta$, which starts from a pre-trained model and is further adapted to a private dataset $\mathcal{D}$. Each text sample $\boldsymbol{x}^{(i)}$ is represented as a sequence of tokens, i.e., $\boldsymbol{x}^{(i)} = (x_1^{(i)}, x_2^{(i)}, ..., x_L^{(i)})$. The sample index $i$ may be omitted for clarity when it is not relevant to the discussion. During inference, the model allows estimating the token likelihood $f_\theta(x_l|x_1, ..., x_{l-1})$ and generates new text by iteratively sampling $\widehat{x}_l \sim f_\theta(x_l|x_1, ..., x_{l-1})$ conditioned on the prefix $(x_1, ..., x_{l-1})$. Starting with the initial token $x_1$, the model feeds each newly sampled token $\widehat{x}_l$ back into itself to generate the subsequent token $\widehat{x}_{l+1}$, continuing this process until a predetermined stopping criterion is met.

**Threat Model.** The attacker $\mathcal{A}$ aims to determine whether a given query text sample was included in the private dataset $\mathcal{D}$ used to customize the target model for the private domain. We adopt the conventional threat model where the attacker may have either *black-box* or *white-box* access to the target model. In the *black-box* scenario, the attacker can access only the model's output probability predictions, typically via a prediction API call. In contrast, the *white-box* scenario permits the attacker to access the model's internal structure and parameters.

We follow the standard evaluation framework, where the adversary has access to a query set $\mathcal{S} = \{(\boldsymbol{x}^{(i)}, m^{(i)})\}_{i=1}^{M}$. This set includes both member (i.e., seen by the target model $f_\theta$) samples and non-member (unseen) samples drawn from the same data distribution. Each $m^{(i)}$ indicates the membership status, where $m^{(i)} = 1$ if $\boldsymbol{x}^{(i)}$ is a member. The attack $\mathcal{A}(\boldsymbol{x}^{(i)}, f_\theta)$ acts as a binary classifier, predicting $m^{(i)}$ for a given query sample $\boldsymbol{x}^{(i)}$ with access to the target model.

### 2.2 Attack Approaches

We conducted a broad literature search to identify representative approaches for membership inference attacks, aiming to provide a comprehensive benchmark. Below, we present an overview of each approach under a unified notation to facilitate comprehension and comparison.

**Likelihood-based** [27]. Given that LLMs are typically trained using a maximum likelihood objective on the training data, the most basic method for predicting membership involves using the (normalized)

log-likelihood of the target query sample as the metric: a *higher* likelihood score indicates a better fit of the target model $f_\theta$ on the query data point $\boldsymbol{x} = (x_1, ..., x_L)$, suggesting it is likely a *member* of the training set. Formally, the attack can be summarized as:

$$\mathcal{A}(\boldsymbol{x}, f_\theta) = \mathbb{1}\Big[\frac{1}{L}\sum_{l=1}^{L} \log f_\theta(x_l|x_1, ..., x_{l-1}) > \tau_L\Big], \tag{1}$$

where $\tau_L$ denotes the threshold score above which the attack predicts the sample to be a member.

**Likelihood with Reference** [12]. While the basic likelihood score provides evidence for membership detection, it often fails to achieve high precision. This is because high-likelihood samples are not always present in the training data, but can also be uninformative texts frequently encountered in the pre-training dataset. A natural improvement involves calibrating the likelihood score by comparing it with the score obtained from a reference model not tailored for the private data. This leads to the likelihood ratio evaluated on the target versus the reference model. Formally,

$$\mathcal{A}(\boldsymbol{x}, f_\theta) = \mathbb{1}\Big[\frac{1}{L}\sum_{l=1}^{L}\Big(\log f_\theta(x_l|x_1, ..., x_{l-1}) - \log f_\phi(x_l|x_1, ..., x_{l-1})\Big) > \tau_{L_{\text{ref}}}\Big], \tag{2}$$

where $f_\phi$ denotes a reference model not trained on the private dataset and $\tau_{L_{\text{ref}}}$ is the threshold.

**Zlib Entropy as Reference** [12]. While using a reference for calibrating the inherent frequency of text is essential for membership inference, it is not necessary to fix the reference to be another neural language model. In principle, any technique that quantifies the normality or informativeness for a given sequence can be useful. Following [12], we compute the zlib entropy of the text, which is the number of bits of entropy when the text sequence is compressed using zlib compression [28]. Subsequently, the ratio of the average negative log-likelihood of a sequence and the zlib entropy is used as the membership inference metric. Formally,

$$\mathcal{A}(\boldsymbol{x}, f_\theta) = \mathbb{1}\Big[-\frac{1}{L}\sum_{l=1}^{L} \log f_\theta(x_l|x_1, ..., x_{l-1})/\mathcal{H}(\boldsymbol{x}) < \tau_{\text{zlip}}\Big], \tag{3}$$

where $\mathcal{H}(\boldsymbol{x})$ denotes the zlib entropy of $\boldsymbol{x}$.

**Neighborhood-based** [29]. To account for the normality of text samples for membership inference, one can calibrate their likelihood scores using their semantic neighbors. This can be achieved by generating neighbors of the data point and measuring their likelihood scores using the target model, which then serve as an estimation for the normality of the query text. The neighbors are designed to preserve semantics and are well-aligned with the context of the original words. These neighbors are obtained through semantically-preserving lexical substitutions proposed by transformer-based masked language models [30]. Formally, the membership score is expressed by comparing the log-likelihood of the query sample to the averaged log-likelihood of its neighbors:

$$\mathcal{A}(\boldsymbol{x}, f_\theta) = \mathbb{1}\Big[\frac{1}{L}\sum_{l=1}^{L} \log f_\theta(x_l|x_1, ..., x_{l-1}) - \frac{1}{kL}\sum_{i=1}^{k}\sum_{l=1}^{L} \log f_\phi(\tilde{x}_l^{(i)}|\tilde{x}_1^{(i)}, ..., \tilde{x}_{l-1}^{(i)}) > \tau_{L_{\text{nbr}}}\Big], \tag{4}$$

where $\{\tilde{\boldsymbol{x}}^{(i)}\}_{i=1}^{k}$ corresponds to $k$ neighbors of the given sample $\boldsymbol{x}$.

**Min-K% Probability** [21]. The MIN-K% Probability score captures the intuition that a non-member example is more likely to include a few outlier words with high negative log-likelihood (or low probability), while a member example is less likely to include words with such low likelihood scores. Following [21], we select the $K\%$ of tokens from $\boldsymbol{x}$ with the minimum token probability to form a set, and compute the average log-likelihood of the tokens in this set

$$\mathcal{A}(\boldsymbol{x}, f_\theta) = \mathbb{1}\Big[\frac{1}{|\text{Min-K\%}(\boldsymbol{x})|}\sum_{x_l \in \text{Min-K\%}(\boldsymbol{x})} \log f_\theta(x_l|x_1, ..., x_{l-1}) > \tau_{\text{Min-K}}\Big], \tag{5}$$

where Min-K%$(\boldsymbol{x})$ denotes the set of tokens with the lowest $K\%$ likelihood conditioned on its prefix.

**Min-K%++** [31]. In the context of maximum likelihood training, it has been observed that training samples tend to form local maxima in the modeled distribution along each input dimension. As

exploring an input dimension can be viewed as substituting the current token with alternative candidates from the model's vocabulary, the membership score is defined by the normalized log probability under the conditional categorical distribution $f_\theta(\cdot|\boldsymbol{x}_{<l})$, where a high probability indicates likely membership. In line with [21], the score is calculated using the Min-K% least probable tokens:

$$\mathcal{A}(\boldsymbol{x}, f_\theta) = \mathbb{1}\left[\frac{1}{|\text{Min-K\%}(\boldsymbol{x})|} \sum_{x_l \in \text{Min-K\%}(\boldsymbol{x})} \frac{\log f_\theta(x_l|x_1,...,x_{l-1}) - \mu_{<l}}{\sigma_{<l}} > \tau_{\text{Min-K++}}\right], \quad (6)$$

while $\mu_{<l} = \mathbb{E}_{z \sim f_\theta(\cdot|\boldsymbol{x}_{<l})}[\log f_\theta(z|\boldsymbol{x}_{<l})]$ represents the expectation of the next token's log probability over the vocabulary of the model given the prefix $\boldsymbol{x}_{<l} = (x_1, ..., x_{l-1})$, and the term $\sigma_{<l} = \sqrt{\mathbb{E}_{z \sim f_\theta(\cdot|\boldsymbol{x}_{<l})}[(\log f_\theta(z|\boldsymbol{x}_{<l}) - \mu_{<l})^2]}$ is the standard deviation.

**Gradient Norm-based** [11]. The phenomenon of local minimality at training data points is often evidenced by the smaller magnitudes of parameter gradients observed at these points [32, 33, 11]. A practical approach would be to utilize the gradient norm of a target data point as the membership score. This concept is mathematically represented as follows:

$$\mathcal{A}(\boldsymbol{x}, f_\theta) = \mathbb{1}\left[\left\| -\frac{1}{L}\sum_{l=1}^{L}\nabla_\theta \log f_\theta(x_l|x_1,...,x_{l-1})\right\| < \tau_{\text{grad}}\right]. \quad (7)$$

Notably, computing this gradient requires white-box access to the target model, unlike the previously mentioned methods, which rely solely on the model's output predictions.

## 3 LLM Adaptation Techniques

Existing LLM adaptation techniques can be roughly categorized into *regular fine-tuning*, *parameter-efficient fine-tuning*, and *in-context learning*. Below, we briefly discuss representative techniques from each of these categories. For a more detailed comparison of parameter-efficient fine-tuning techniques, we refer readers to prior work [34].

**Regular Fine-tuning**. The basic fine-tuning approach involves taking a pre-trained model and adapting all its parameters for a task-specific downstream dataset, i.e., *full fine-tuning*. This enables the model to learn specific patterns in the new data domain, thereby improving its accuracy and relevance for the target application. However, as models increase in size, full fine-tuning becomes impractical due to the high computational cost. Additionally, overfitting can become a significant issue, closely related to privacy vulnerabilities.

**Adapter**. Adapter-based fine-tuning strategically integrates additional lightweight layers into an existing model architecture [35, 36, 37], typically by injecting small modules (adapters) between transformer layers. During fine-tuning, only these adapter layers are updated for domain-specific data, while the core model parameters remain frozen, which greatly reduces computational overhead compared to regular fine-tuning.

**Low-Rank Adaptation**. Low-Rank Adaptation (LoRA) [38] is based on the hypothesis that weight changes during model adaptation exhibit a low "intrinsic rank". To leverage this, LoRA proposes integrating trainable low-rank decomposition matrices into each transformer layer to approximate the weight updates, while only allowing modifications of these low-rank matrices and freezing the pre-trained weights.

**Prompt-based Tuning**. Instead of changing the weights of the neural network, prompt-based tuning [39] typically involves adding specific prompts to the input text to steer the model towards the desired output. Existing studies commonly prepend tunable continuous task-specific vectors to the input embeddings (potentially across multiple layers), typically known as "soft prompts", and optimize over these continuous prompts while keeping the other pre-trained parameters unchanged during the fine-tuning process. Specifically, *Prompt-tuning* [40] prepends the input sequence with special tokens to form a template and tune the embeddings of these tokens directly. *P-tuning* [41] adds continuous prompt embeddings generated from pseudo prompts by a small encoder to the input embeddings of the model and tunes the prompt encoder. *Prefix tuning* [42] injects a trainable prefix matrix into the keys and values of the multihead attention at every layer of the model and updates the injected trainable prefix matrices.

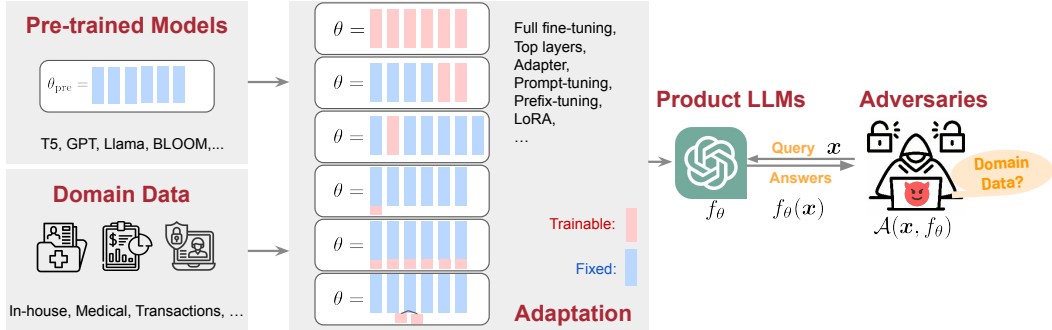

Figure 1: An overview pipeline illustrating the workflow of `PrivacyAuditor`.

**In-context Learning**. By enabling LLMs to perform diverse tasks through contextual adaptation, without altering their internal parameters, in-context learning [43] introduces a paradigm shift from traditional fine-tuning. Instead of performing explicit parameter updates, the model utilizes task-specific examples and instructions embedded within the input prompt to infer the task requirements. The key insight lies in the model's ability to treat these examples as implicit demonstrations, dynamically aligning its behavior with the desired output. This emergent capability makes in-context learning highly flexible, as it allows the model to generalize effectively from limited examples with minimal computational overhead, avoiding the computational burden associated with fine-tuning [44].

## 4 Related Work

**Privacy Threat for LLMs**. While the rapid development of LLMs has greatly facilitated various real-world applications, the widespread use of LLMs, especially in sensitive domains such as medical and finance, has raised serious privacy concerns. It is notorious that large neural networks tend to unintentionally memorize their training data (beyond learning the general patterns essential for conducting the target tasks), which raises vulnerabilities to privacy attacks such as membership inference [19, 7, 8, 9, 18, 21, 29, 45, 46, 47, 48, 49, 50, 51, 52], personal identifiable information retrieval [13, 14, 53, 54], and training data extraction [11, 12, 15, 53].

**Membership Inference in LLMs**. Membership inference is a commonly studied privacy attack, which is closely related to other topics such as training data extraction (by serving as an intermediate step) [12], examining data contamination [21] (i.e., whether the testing data have been seen by the target model), and theoretical privacy notions like differential privacy [20] (which by construction should provide privacy guarantees in the context of training data membership). While recent studies have investigated such attacks for data used for model pre-training [46, 21, 50, 51, 52, 55] and fine-tuning [7, 8, 9, 10, 11], they are focusing on specific attack strategies, a limited set of fine-tuning techniques (typically full fine-tuning or tuning the top layers) and particular model types (e.g., pre-trained encoders), which may not faithfully reflect the existing progress of such investigation.

To address this gap, our work considers a broad range of representative recent adaptation techniques and attack methods. This includes literature that may not directly focus on membership inference but is applicable to it. Our investigation aims to provide a more comprehensive understanding of potential privacy threats related to membership leakage when using LLMs.

## 5 Experiments

### 5.1 Setup

**Datasets**. In contrast to previous studies, which have primarily focused on less sensitive datasets such as News and Wikipedia, our study is dedicated to a detailed evaluation of private data leakage risks in environments that handle highly sensitive and valuable private information. Specifically, we conduct experiments on the following adaptation datasets $\mathcal{D}$: Sujet-finance-instruct-177k (**Suject Finance**) [56], Corporate Climate Policy Engagement (**CorpClimate**) [57], as well as Synthetic-Text-to-SQL (**SQL**) [58]. Our selection process aimed to minimize potential overlap with the pre-training

datasets and ensure a more accurate evaluation of membership. Specifically, all the chosen fine-tuning datasets were released after the pre-trained models were developed, reducing the risk of shared content. Additionally, the datasets underwent extensive pre-processing to further minimize the chance of overlapping data points, even if they might originate from similar sources. We also included synthetic data with a specific structure that is unlikely to derive from web-based sources, ensuring further independence from the data used in pre-training.

**Models**. We consider the two predominant LLM architectures: decoder-only and encoder-decoder LLMs and conduct experiments on foundation models including **T5** [3], **LLaMA** [22], **OPT** [23], **BLOOM** [24], and **GPT-J** [25], each configured with different numbers of model parameters. All the open-source pre-trained LLMs are downloaded from Huggingface[1]. All experiments are conducted on a computing cluster with 4 Nvidia A100 80G with 512G memory. More details are included in the supplementary materials.

**Evaluation Configuration**. We evaluate the target LLMs' test accuracy on the test portion of the adaptation datasets as the *utility* metric. For evaluating privacy, following the common evaluation standard for membership inference attacks, we composed an evaluation query set $S$ comprising an equal number of member and non-member samples (defaulting to 1000 each), while limiting the sample size to 10 for in-context learning experiments due to memory constraints. The member samples are uniformly sampled from the training dataset, while the non-member samples are randomly selected from the test portion of the datasets, ensuring they were not used in training. Privacy leakage is evaluated using standard metrics [46], including attack Area under the ROC Curve (**AUC-ROC**), False Positive Rate at low True Positive Rate (**FPR@0.1%TPR**, and **FPR@1%TPR**).

**Attack and Adaptation Techniques**. We evaluate the following attack methods as outlined in Section 2.2: **Likelihood** (Equation 1), **Likelihood-ref** (Equation 2), **Zlib Entropy** (Equation 3), **Neighborhood** (Equation 4), **Min-K** (Equation 5), **Min-K++** (Equation 6), **Gradient-Norm** (Equation 7) as outlined in Section 2.2. As introduced in Section 3, we evaluate the following representative adaptation techniques: full fine-tuning (**Full**), only updating the attention heads of the top-2 layers (**Top2Head-tuning**), adapter-based technique (**Adapter-H** [35]), **Prefix-tuning** [42], **LoRA** [38], **P-tuning** [41], **Prompt-tuning** [40], and **in-context learning** [43]. Note that all the aforementioned attack methods require black-box access to the target model, except for the Gradient-Norm method. This exception may render the Gradient-Norm method inapplicable to typical in-context learning scenarios where no parameter updates are performed. We use the default parameters from the original implementations. More details can be found in the supplementary materials.

## 5.2 Benchmark Design

To systematically assess data leakage risks across various fine-tuning approaches in LLMs, we present experiments designed to answer the following research questions.

### RQ1: Is Private Data Used for Adapting LLMs Vulnerable to Leaks?

**Motivation.** Although LLMs demonstrate promising capabilities in generalizing across multiple tasks, adapting them to specific domain applications remains essential due to non-negligible domain shifts [59]. Since domain data is a crucial asset for data owners and typically contains sensitive information, it is vital to assess the extent to which this data can be leaked from the product model.

**Approach.** We first adopt the arguably most competitive lightweight fine-tuning technique, namely LoRA, to generate target downstream models across different datasets. Then, we visualize the data distributions of the member and non-member likelihood scores and inspect whether systematic differences exist that can be used as clues for detecting membership. Subsequently, we employ various state-of-the-art MIAs to measure the extent of private domain information leakage.

### RQ2: Do Different Adaptation Techniques Vary in Their Downstream Privacy Vulnerability?

**Motivation.** Different adaptation techniques involve distinct design patterns, introduce varying computational costs, and achieve unequal target performance. While these aspects have been extensively compared in existing literature on (parameter-efficient) fine-tuning techniques, the corresponding privacy implications have not been thoroughly investigated. Therefore, we design experiments to examine how various adaptation methods affect the effectiveness of privacy attacks.

---

[1]`https://huggingface.co/models`

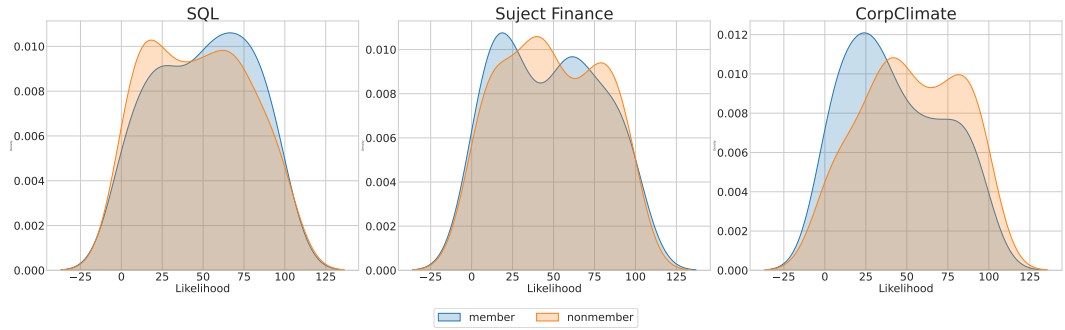

Figure 2: The likelihood score distribution of member and non-member data in Llama-7b fine-tuned with LoRA on different datasets.

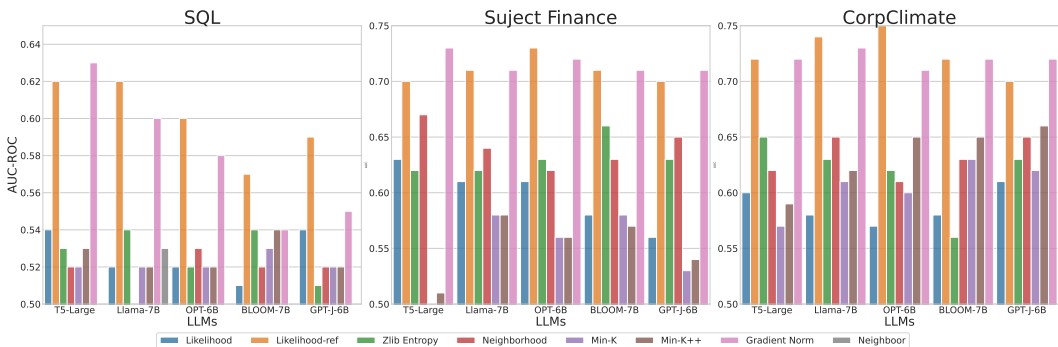

Figure 3: Overview of the attack performance across different LLMs and datasets.

**Approach.** We provide a unified implementation of representative adaptation techniques with varying amounts of trainable parameters. We then compare the performance of MIAs and model utility across various datasets and evaluation metrics under fair comparison conditions.

**RQ3: What Factors Potentially Affect Privacy Vulnerability in LLM Adaptation?**

**Motivation.** Besides knowing *"whether"* different LLM adaptation techniques affect the privacy vulnerability of the resulting product LLM, it is also crucial to understand *"how"* and *"why"*. Investigating the potential factors that influence such vulnerability is essential, as understanding these factors is beneficial for developing more robust and privacy-preserving LLM fine-tuning approaches, and provides insights into preventing private domain data from leaking during the fine-tuning process.

**Approach.** Motivated by the existing understanding of privacy risks associated with large neural networks, we conduct experiments spanning several critical factors: varying amounts of data for adaptation, different numbers of training iterations, and various model sizes. Additionally, we perform fine-tuning on domain datasets for both multiple tasks and single tasks, aiming to examine how task diversity in the pre-training dataset affects privacy vulnerability.

### 5.3 RQ1: Is Private Data Used for Adapting LLMs Vulnerable to Leaks?

**Distributional Differences Between Member and Non-Member Data.** Figure 2 visualizes the distribution of likelihood scores for member and non-member data using the target *Llama7b* model fine-tuned with *LoRA*. Even though these likelihood scores (Equation 1) represent the most basic metric an attack would consider, the results reveal subtle but noticeable distinctions in the distributions. This indicates the potential for an adversary to exploit LLM outputs to determine whether a sample was used in fine-tuning and highlights the vulnerability of membership leakage of domain data through deployed product LLMs. However, the limited prominence of these differences also underscores the need for more refined attack strategies to effectively uncover membership information.

**Strong MIAs Effectively Detect Data Used for LLM Adaptation**. Given the distinct distribution patterns between member and non-member data, we conducted experiments on existing representative

```
DELETE FROM space_debris WHERE weight < 100;          DELETE FROM Museums WHERE Attendance < 5000
DELETE FROM farmers WHERE age > 60;                   DELETE FROM policies WHERE state = 'Texas';
DELETE FROM cultural_sites WHERE site_id = 2;         DELETE FROM space_debris WHERE diameter < 10;
DELETE FROM tv_shows WHERE genre = 'Horror';          DELETE FROM Public_Services WHERE service_id = 2;
```

(a) Member Data                           (b) Misclassified Nonmember Data

Figure 4: The comparison of samples between member data and misclassified non-member data from Llama7b fine-tuned over the SQL dataset using LoRA. We apply reference-based MIA [12] to conduct the membership inference attack.

(a) T5-Large

| Adaptation Method | Attack Method | | | | | | | Accuracy (after) |
|---|---|---|---|---|---|---|---|---|
| | Likelihood | Likelihood-ref | Zlib Entropy | Neighborhood | Min-K | Min-K++ | Gradient-Norm | |
| Prompt-tuning | 0.567 | 0.609 | 0.572 | 0.582 | 0.544 | 0.549 | 0.621 | 0.631 |
| Prefix-tuning | 0.589 | 0.626 | 0.621 | 0.606 | 0.585 | 0.592 | 0.644 | 0.637 |
| Adapter-H | 0.574 | 0.691 | 0.597 | 0.611 | 0.552 | 0.556 | 0.696 | 0.639 |
| P-tuning | 0.591 | 0.694 | 0.614 | 0.619 | 0.579 | 0.583 | 0.707 | 0.623 |
| LoRA | 0.592 | 0.724 | 0.647 | 0.624 | 0.567 | 0.588 | 0.717 | 0.644 |
| Top2-head | 0.623 | 0.726 | 0.658 | 0.631 | 0.584 | 0.593 | 0.733 | 0.637 |
| Full | 0.817 | 0.853 | 0.831 | 0.811 | 0.822 | 0.825 | 0.858 | 0.643 |
| In-Context | 0.881 | 0.881 | 0.881 | 0.881 | 0.881 | 0.881 | 0.881 | 0.458 |
| From scratch | 0.887 | 0.943 | 0.914 | 0.909 | 0.892 | 0.921 | 0.958 | 0.604 |

(b) Llama-7B

| Adaptation Method | Attack Method | | | | | | | Accuracy (after) |
|---|---|---|---|---|---|---|---|---|
| | Likelihood | Likelihood-ref | Zlib Entropy | Neighborhood | Min-K | Min-K++ | Gradient-Norm | |
| Prompt-tuning | 0.562 | 0.629 | 0.591 | 0.619 | 0.554 | 0.579 | 0.635 | 0.664 |
| P-tuning | 0.587 | 0.636 | 0.628 | 0.633 | 0.583 | 0.595 | 0.644 | 0.676 |
| Prefix-tuning | 0.574 | 0.648 | 0.633 | 0.635 | 0.577 | 0.601 | 0.642 | 0.671 |
| Adapter-H | 0.556 | 0.675 | 0.607 | 0.628 | 0.566 | 0.579 | 0.659 | 0.669 |
| LoRA | 0.575 | 0.735 | 0.634 | 0.654 | 0.608 | 0.622 | 0.728 | 0.674 |
| Top2-head | 0.677 | 0.788 | 0.714 | 0.694 | 0.647 | 0.696 | 0.793 | 0.669 |
| Full | 0.832 | 0.882 | 0.847 | 0.803 | 0.787 | 0.827 | 0.879 | 0.677 |
| In-Context | 0.922 | 0.922 | 0.922 | 0.922 | 0.922 | 0.922 | 0.922 | 0.534 |
| From scratch | 0.913 | 0.943 | 0.914 | 0.899 | 0.892 | 0.921 | 0.958 | 0.278 |

Table 1: Comparison of different adaptation techniques in terms of attack vulnerability (measured by AUC-ROC) and downstream utility (evaluated by model accuracy *after* adaptation) on the T5-Large/Llama-7B model and CorpClimate dataset. The adaptation methods are sorted by ascending order in terms of the amounts of trainable parameters. The shaded area indicates the reference results from training the model from scratch. For reference, the baseline test accuracy *before* adaptation is 0.334 (pre-trained) or 0.187 (from scratch) for the T5-Large model, and 0.493 (pre-trained) or 0.234 (from scratch) for the Llama-7B model.

MIAs (outlined in Section 2.2) to determine whether these differences can be exploited to infer the membership of a given sample. As summarized in Figure 3, the results demonstrate that LLM adaptation techniques may lead to the leakage of training data under existing attacks, with *Likelihood-ref* (Equation 2) being the most effective method overall and performing reasonably well across different types of model architectures. These results represent a meaningful lower bound on the worst-case privacy risk, highlighting the privacy vulnerabilities introduced during LLM fine-tuning and underscoring significant data protection demands during LLM fine-tuning. The complete quantitative results are presented in the supplementary materials.

**Product LLMs for Structural Data Demonstrate Greater Robustness Against MIAs.** As shown in Figure 3, inferring membership on the SQL dataset is more difficult than on the others. This may be due to the structural similarity of data samples within the same distribution, i.e., smaller in-domain diversity. To validate this, we further analyze the data samples misclassified by the attacker (shown in Figure 4) and observe that these data are structurally identical and semantically highly similar. This may indicate a current weakness in attack methods that rely on detecting individual patterns or fingerprints (which are largely based on semantics and structure) memorized by the target model.

## 5.4  RQ2. The Impact of Adaptation Techniques on Downstream Privacy Vulnerability.

**More Trainable Parameters Lead to Higher Data Membership Leakage Risk**. Figures 5 & 6 offer an overall performance comparison of different adaptation techniques on the adapted *OPT-6b*

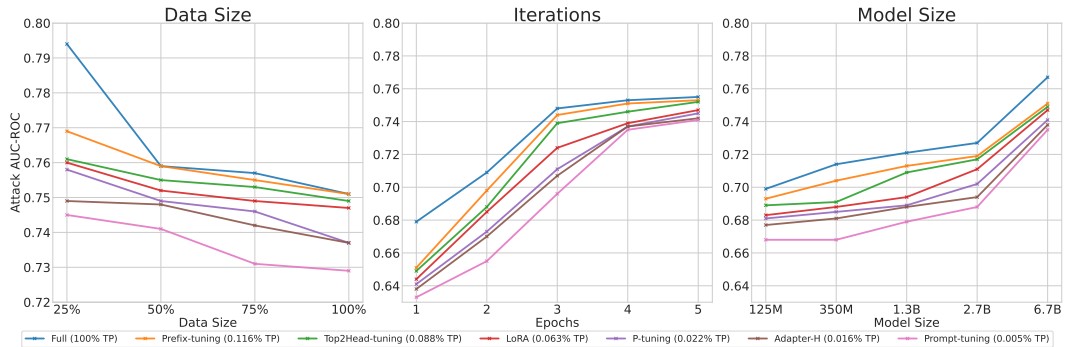

Figure 5: Impact of different adaptation techniques for *attack performance* measured by AUC-ROC. TP refers to the percentage of trainable parameters compared to the full-size model parameters.

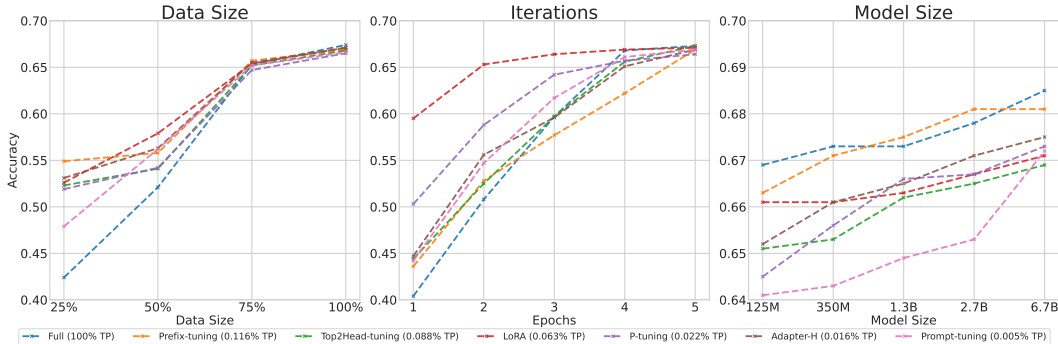

Figure 6: Impact of different adaptation techniques for *model utility* measured by accuracy. TP refers to the percentage of trainable parameters compared to the full-size model parameters.

model for the *CorpClimate* dataset. The portion of trainable parameters (*TP*) relative to the overall model size is listed in brackets beside each adaptation technique, with techniques ordered in the legend by decreasing trainable parameters. The results show that the more parameters applied during adaptation, the higher the risks of downstream membership leakage. This aligns with the intuition that models with more trainable parameters tend to have a higher degree of freedom in downstream adaptation, potentially allocating more modeling capacity to over-memorizing their training data. While in-context learning approaches do not involve parameter updates and thus avoid the same overfitting risks, they are not free of privacy concerns. As shown by the non-trivial attack performance in Table 1, training data embedded within the language model through in-context adaptation can potentially be extracted through careful analysis of model outputs. This suggests that even parameter-free techniques require careful monitoring of the risk of privacy leakage.

**Different Adaptation Techniques May Cause Systematic Vulnerability Differences Due to Their Associated Attack Surfaces**. As illustrated in Table 1, different adaptation methods exhibit varying degrees of vulnerability to attack methods (measured by AUC-ROC) and post-adaptation utility (evaluated by accuracy). Specifically, adaptation techniques can introduce varying attack surfaces influenced by factors beyond the size of trainable parameters, such as the degree of model modification, the layers involved, and practical usage scenarios. For instance, methods like prompt-tuning and P-tuning primarily adjust input representations, potentially reducing the attack surface but offering moderate performance gains. In contrast, approaches like LoRA or full fine-tuning modify deeper layers, which may enhance flexibility but also increase the chances of embedding sensitive information within parameters. In-context learning, which relies on input data at runtime without parameter updates, is typically employed in black-box settings, where attackers have limited access to model internals, making white-box attack assumptions less applicable. These differences emphasize the importance of aligning adaptation techniques with both performance needs and privacy considerations.

### 5.5 RQ3. Factors Affecting Privacy Vulnerability.

**Size of Domain Data Applied for Training.** Figure 5 demonstrates the empirical assessment of privacy leakage risks with varying amounts of available data for LLM adaptation. Utilizing more data tends to shift the LLM's modeling capability towards generalization rather than specialization, leaving less room for it to overfit to individual patterns, thus making the attack less effective. Moreover, using more data samples aligns with the utility objectives of product LLMs, as shown in Figure 6, which suggests the necessity of always obtaining more data for training.

**Number of Fine-tuning Iterations.** As can be observed from Figure 5, increasing the number of iterations generally enhances the effectiveness of attacks on the target models. This aligns with the interpretation that a higher degree of adaptation to the domain data, while steering the LLMs towards the target domain, inevitably causes the model to learn patterns overly tailored to individuals rather than the essential ones required for the task. While the privacy objective suggests applying a lesser degree of adaptation to the domain data, the utility objectives of product LLMs require a high degree of fitting to the target domain data. This misalignment of objectives necessitates more detailed adjustments during the deployment phase.

**Target Model Size.** From Figure 5, we observe that larger LLMs tend to exhibit increased downstream privacy vulnerability after adaptation. This may be attributed to their greater model capacity, which, while enabling the learning of more complex patterns and solving difficult tasks, can also compromise individual privacy, as the enhanced capacity allows these models to learn personal information that can lead to privacy issues. This dilemma between learnability (and thus utility) and privacy also requires more dedicated efforts for adjustments during the deployment phase.

## 6 Discussion & Limitations

While our results offer valuable insights into privacy-aware LLM development, several areas remain open for further exploration to deepen this research. One important direction is studying the impact of privacy-preserving training mechanisms, such as differentially private adaptation, which, while offering theoretical guarantees, may introduce utility trade-offs, particularly for complex tasks like domain-specific reasoning. Understanding how such strategies influence both membership inference risks and model utility, along with their trade-offs, is crucial for guiding practitioners. Another promising avenue is the co-design of privacy-preserving techniques with efficient adaptation methods, as developing these independently can result in suboptimal outcomes. An integrated approach may better balance privacy and utility, and identifying inherently robust adaptation techniques could reduce the need for costly post-hoc defenses. Additionally, auditing tools that search for or generate vulnerable samples could provide more precise estimates of privacy leakage and support ongoing monitoring of deployed models to maintain an appropriate privacy-utility balance.

Finally, it is essential to acknowledge the limitations of this work. While the evaluation focuses on domains intended to reflect real-world scenarios, it may not capture the full range of potential attack settings. Attackers with specialized knowledge or additional assumptions could uncover vulnerabilities beyond those examined. Moreover, the privacy risks identified are bound by the framework used, with results varying across datasets, model architectures, and operational contexts. Future work could expand this benchmark by incorporating new adaptation techniques, datasets, and attack strategies, progressively advancing the understanding of privacy risks across diverse settings.

## 7 Conclusions

In this work, we present a benchmark to assess the potential privacy leakage risks during adaptation techniques in LLMs. We examine the training data membership leakage risk in mainstream large language models based on encoder-decoder and decoder-only structures. Our comprehensive analysis illustrates the facets of privacy leakage risks during LLM adaptation, and we further propose a unified platform to measure these potential privacy risks. Our findings highlight the importance of developing privacy-preserving adaptation techniques with practical relevance.

## Acknowledgments

We thank the anonymous reviewers for their valuable comments and constructive feedback, which have significantly improved the quality of this work. This work is supported in part by Canada CIFAR AI Chairs Program, the Natural Sciences and Engineering Research Council of Canada (NSERC No.RGPIN-2021-02549, No.RGPAS-2021-00034, No.DGECR-2021-00019); as well as JST-Mirai Program Grant No.JPMJMI20B8, JSPS KAKENHI Grant No.JP21H04877, No.JP23H03372, No.JP24K02920, and also with support from TIER IV, Inc., and the Autoware Foundation.

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

# Supplementary materials

These supplementary materials provide detailed information on the experimental setup (see §A) and present additional results (see §B). The source code implementation can be accessed via the following link: https://github.com/sunshine-collab/PrivAuditor.

## A Experiment Setup

### A.1 Dataset

**Sujet Finance Dataset** [56][2]. The Sujet Finance dataset is a comprehensive collection of financial data crafted specifically for fine-tuning LLMs for specialized financial tasks. It aggregates data from 18 distinct HuggingFace datasets, comprising 177,597 entries across seven key financial LLM tasks: sentiment analysis (44,209 entries), direct question answering (38,801 entries), question answering with context (40,475 entries), conversational question answering (15,613 entries), yes/no questions (20,547 entries), topic classification (16,990 entries), and entity-level sentiment analysis (962 entries). The data record is structured with columns such as inputs, answers, system prompts, user prompts, dataset names, task types, index levels, and conversation IDs. The dataset undergoes de-duplication and preprocessing to eliminate non-ASCII and other irregular characters, making it a clean and usable dataset for effective LLM fine-tuning. We fine-tune the LLMs on all tasks contained in the dataset and evaluate the model utility on classification tasks (including "Sentiment Analysis", "Yes/No Questions", "Topic Classification", and "NER Sentiment Analysis") that allow easy quantification using accuracy. The query sample $x$ corresponds to the complete input to the model, which comprises an "instruction" combined with an "input". See Table 2 for examples.

```
instruction : You are a financial analyst categorizing tweets into specific financial topics. Given a tweet, classify it into one of the
              following topics: ['Analyst Update', 'Fed | Central Banks', 'Company | Product News', 'Treasuries | Corporate Debt', 'Dividend',
              'Earnings', 'Energy | Oil', 'Financials', 'Currencies', 'General News | Opinion', 'Gold | Metals | Materials', 'IPO', 'Legal |
              Regulation', 'M&A | Investments', 'Macro', 'Markets', 'Politics', 'Personnel Change', 'Stock Commentary', 'Stock Movement'].
              Output only the topic name.
input : Goldmans economists see a 50% chance of a recession in the next 24 months, CEO David Solomon tells @kaylatausche.
        https://t.co/bHzTr7ZVUm https://t.co/kdYhTnQCzf
output : Macro
answer : Macro
```

```
instruction : As a finance expert, your role is to provide insightful answers to questions related to finance. Given a specific request and
              relevant context, carefully analyze the information provided and formulate a detailed, accurate response that addresses the key
              points of the request.
input : Request:\nConstruct a logical introduction for the essay.\n\nContext:\nEssay:\n\nJohn Doe is a chef and a true innovator in the food
        industry. He has been passionate about cooking since he was a child and has always been looking for ways to introduce unique and
        exquisite dishes to his diners. He draws his inspiration from his childhood memories and experiments with different flavors and
        ingredients to create surprisingly delightful dishes. As a result, John Doe has become a go-to chef for many top restaurants and
        specialty stores across the country.
output : Introduction:\nThe culinary landscape is constantly evolving, driven by the imagination and ingenuity of chefs who push the
         boundaries of flavor and presentation. Among them stands John Doe, a trailblazer in the food industry. His unwavering passion for
         cooking, honed by years of experience, has led him on a remarkable journey of culinary exploration and innovation. From his childhood
         fascination with the art of food to his groundbreaking dishes that delight diners across the nation, John Doe's remarkable journey is
         a testament to the power of creativity and dedication in the pursuit of culinary excellence.
answer : John Doe is a chef whose dedication to creativity and innovation in the food industry has earned him widespread recognition. With
         passion for cooking stemming from his childhood days, he has gone on to explore different flavors and ingredients to bring unique and
         exquisite dishes to his diners. As a result, he has become the go-to chef for many top restaurants and specialty stores across the
         country.
```

```
instruction : As an experienced financial advisor, your role is to provide insightful and accurate responses to financial questions and
              topics. Leverage your extensive knowledge in personal finance, financial markets, and public sentiment to offer well-informed
              guidance. Consider the context, if provided, to frame your answer.
input : Question about Personal Finance:\nHow did you manage to save $10,000 while working as a truck driver and still pay off your debt?
        \n\nContext:\nThe original post was a Reddit post from someone who saved $10,000 in savings for the first time in their life and
        shared their story, including their job as a truck driver and how they used Dave Ramsey's method to pay off their debt.
output : Remember: Debt repayment is a journey, not a destination. Stay motivated, celebrate milestones, and don't be discouraged by setbacks.
answer : I reached my goal of $10,000 in savings for the first time in my life by working hard and using Dave Ramsey's 7-step method to pay
         off my debt. I've been working almost exclusively 70-hour weeks for six months to get this far, and I've been using the money I've
         saved to pay off my outstanding bills from a long time ago.
```

```
instruction : You are a financial sentiment analysis expert. Your task is to analyze the sentiment expressed in the given financial text.Only
              reply with positive, neutral, or negative.
input : Central bank chief Pan Gongsheng has pledged to keep monetary policy accommodative to support the post-pandemic recovery, but also
        urged structural reforms to reduce reliance on infrastructure and property for growth.
output : Neutral. The text expresses a balancing tone, acknowledging the need for both monetary stimulus and structural reforms to sustain
         economic growth.
answer : negative
```

Table 2: Examples of Sujet Finance Dataset Records. Each query sample consists of an "instruction" concatenated with an "input", while the "answer" represents the ground-truth label of the dataset. The "output" is a demonstration of the LLM's response to the query sample.

---

[2]https://huggingface.co/datasets/sujet-ai/Sujet-Finance-Instruct-177k

**Corporate Climate Policy Engagement** [57][3]. The dataset is designed to estimate corporate climate policy engagement by analyzing various PDF-formatted documents derived from LobbyMap. It includes 11,159 documents annotated for corporate stances on climate policies. Each document's text is extracted and organized into triplets $(P, Q, S)$, where $Q$ represents high-level climate policy issues, $S$ denotes the stance on a five-level scale from "strongly supporting" to "opposing", and $P$ indicates the evidence page indices supporting the query and stance. The dataset is provided in JSON format with fields such as *document ID*, *sentences* (including sentence ID and page numbers for task input), *evidences* (containing $P$, $Q$, and $S$), and *meta* (offering additional metadata about the evidence items). Preprocessing involved robust text extraction using tools like docTR, Tesseract, and PyMuPDF, OCR for necessary alignment, de-duplication, and data cleaning to ensure quality. See Table 3 for examples of the dataset.

---

instruction : What is the stance of the corporate climate policy engagement for \"ghg_emission_regulation\" with the given statement? Answer in one of the following 5 options: no_position_or_mixed_position, not_supporting, opposing, strongly_supporting, supporting. \n

input : Statement: Home / Models / BMW i / BMW CEO Krueger: EU's 2030 CO2 target is… BMW CEO Krueger: EU's 2030 CO2 target is unattainable  BMW i | October 6th, 2018 by Horatiu Boeriu 17 comments Tweet Like 4  Save  At the Paris Motor Show, BMW CEO Harald Krueger said a higher reduction of CO2 emissions foreseen in Europe is simply unattainable.  "Hoping to reduce …  At the Paris Motor Show, BMW CEO Harald Krueger said a higher reduction of CO2 emissions foreseen in Europe is simply unattainable.  "Hoping to reduce CO2 emission by 45 percent by 2030 is dreaming.  It is just not possible," Krueger said.  Last week, European Union lawmakers voted to impose a slightly lower CO2 limit of 40 percent by 2030, stricter than initial proposals of 30 percent.  "To get to a 45 percent CO2 reduction, we would need 70 percent of European sales being battery-powered vehicles, and the power infrastructure simply would not be able to handle it," Krueger said.  Settings  This website uses cookies  We use cookies to ensure that we give you the best experience on our website.  This includes cookies from third party social media websites if you visit a page which contains embedded content from social media.  Such third party cookies may track your use of the BMWBLOG website.  We and our partners also use cookies to ensure we show you advertising that is relevant to you.  If you continue without changing your settings, we'll assume that you are happy to receive all cookies on the BMWBLOG website.  However, you can change your cookie settings at any time.  OK 

output : The statement expresses the CEO's belief that the EU's 2030 CO2 target is unattainable, suggesting opposition to the regulation.

correct_answer : opposing

---

instruction : What is the stance of the corporate climate policy engagement for \"energy_transition_&_zero_carbon_technologies\" with the given statement? Answer in one of the following 5 options: no_position_or_mixed_position, not_supporting, opposing, strongly_supporting, supporting. \n

input : Statement: 25/08/2022, 17:01 CEO Alfred Stern on the OMV Strategy 2030  What will decide the long-term success in the implementation of the OMV Strategy 2030?  First and foremost, of course, the decisive factor will be our ability as a company to make this strategy a re- ality.  As | said, |am convinced that we are in a first-class position here.  We have many decades of experi- ence, extremely competent employees, we are active worldwide, offer the best products in a lot of market segments and have first-class partner companies.  This is a very good basis for our plan to become a leading supplier of sustainable fuels, chemicals and materials by 2030 and to transform our value chain to a circular economy.  A priority will be consistently pursuing our targets and further deepening cooperation at every level across the board as it is important to jointly utilize all available potential.  What external factors do you consider essential?  A strategy cannot be developed independently of the political framework and the market environment.  Particularly in an area of climate protection, it will also require a corresponding regulatory framework in order to bring new technologies forward in a timely fashion.  Take Carbon Capture & Storage, for example, which is currently one of the best technologies for reducing global greenhouse gas emissions, but for which there is still no legal basis in some countries.  However, changing the way society as a whole thinks and acts will be important too.  We also need to focus more on cooperation. i.e.  Within the economy, i.e.  partner compa- nies and other industry participants, but also between business, academia and politics.  And we must ensure that there is sufficient demand for sustainable products, because if sustainable products and solutions are not taken up, then there is no point in offering them long term.  More information: OMV Strategy 2030: From Value Chain to Value Circle Press release: OMV Strategy 2030: Fundamental shift from linear towards circular business approach  Tags: Circulareconomy.  _– Strategy  Related content  https://;www.omv.com/en/blog/ceo-alfred-stern-on-the-omv-strategy-2030 3/4 

output : Stance on corporate climate policy engagement: Supporting, with a focus on regulatory frameworks and market collaboration

correct_answer : no_position_or_mixed_position

---

instruction : What is the stance of the corporate climate policy engagement for \"alignment_with_ipcc_on_climate_action\" with the given statement? Answer in one of the following 5 options: no_position_or_mixed_position, not_supporting, opposing, strongly_supporting, supporting. \n

input : Statement: February 9, 2022  The Honorable Charles E. Schumer The Honorable Nancy Pelosi  Majority Leader Speaker  U.S. Senate U.S. House of Representatives  Washington, D.C. 20510 Washington, D.C. 20515  Dear Leader Schumer and Speaker Pelosi:  As leading companies from a range of sectors across the U.S. economy, we believe that ambitious climate action is a business imperative.  Meeting this global challenge will require bold and timely leadership from federal policymakers, and we thank you for your work in advancing strong climate provisions as part of the Build Back Better framework.  As negotiations on Congress' legislative priorities proceed, we urge you to work to overcome the present impasse and see these historic climate and clean energy investments are realized.  Their enactment would not only help solidify America's global leadership in addressing the climate crisis, but also create a foundation for the long- term prosperity and resilience of communities across the United States.  America's ability to compete in a low-carbon global economy will be shaped by the choices we make today.  As leaders in our industries, we are committed to tackling the climate crisis and are making significant investments of our own to reduce emissions and create the low- and net-zero carbon products and services that will power the global economy in the decades to come.  The actions you take to invest in U.S. leadership in the low-carbon economy will greatly affect the extent to which we can realize the commercial opportunities associated with the export of technologies, products and expertise.  The climate and clean energy provisions in Build Back Better, including tax credits for innovation as well as grants and other funding to support communities in transition, would harness market forces and help spur private sector investment at the scale needed to meet our long-term climate goals.  Crucially, these investments will also support the growth of sustainable domestic industries and the good jobs that come with them in communities across the country.  Last year alone, the U.S. experienced 20 weather and climate events exceeding $1 billion in costs, resulting in more than $145 billion in losses.  As the human and economic costs of catastrophic wildfires, flooding and hurricanes, and other extreme weather continue to grow, bold and timely action is critical.  In addition to their economic benefits, the investments spurred by the climate and clean energy provisions in Build Back Better will play a critical role in meeting our nation's commitments under the Paris Agreement, including our 2030 nationally determined contribution.  U.S. leadership is an indispensable part of a net-zero future, and we simply should not wait any longer to take meaningful action to address climate change. 

output : Stance on alignment with IPCC on climate action: Strongly supporting

correct answer : strongly supporting

---

Table 3: Examples of Corporate Climate Policy Engagement Records. Each query sample consists of an "instruction" concatenated with an "input", while the "correct_answer" represents the ground-truth label of the dataset. The "output" is a demonstration of the LLM's response to the query sample.

**Syntatic-Text-to-SQL** [58][4]. This dataset, generated by Gretel Navigator, is designed to train models for translating natural language into SQL queries. It includes around 105,851 entries, totaling

---

[3] https://climate-nlp.github.io/

[4] https://huggingface.co/datasets/gretelai/synthetic_text_to_sql

approximately 23 million tokens, of which 12 million are SQL-specific. It spans 100 distinct domains or verticals and encompasses a comprehensive suite of SQL tasks, including data definition, retrieval, manipulation, analytics, and reporting. Each features attributes such as SQL complexity, task type, and domain descriptions. The dataset is structured in JSON format with fields for document IDs, tokenized text, and SQL queries. Preprocessing involves text extraction, OCR for alignment, and data cleaning. The default training dataset size is set to be 60,000. See Table 4 for examples of the dataset.

```
instruction : Insert a new record in the 'Faculty_Members' table with the following details: Faculty_ID = 20, First_Name = 'Sofia', Last_Name
              = 'Ahmed', Title = 'Professor', Department = 'Computer Science', Hire_Date = '2018-01-01', Salary = 85000
input : value
output : INSERT INTO Faculty_Members (Faculty_ID, First_Name, Last_Name, Title, Department, Hire_Date, Salary) VALUES (20, 'Sofia', 'Ahmed',
              'Professor', 'Computer Science', '2018-01-01', 85000);
correct_answer : INSERT INTO Faculty_Members (Faculty_ID, First_Name, Last_Name, Title, Department, Hire_Date, Salary) VALUES (20, 'Sofia',
              'Ahmed', 'Professor', 'Computer Science', '2018-01-01', 85000);
```
```
instruction : Insert a new digital asset with the name 'CryptoPet', symbol 'CPT', and total supply of 1,000,000,000 into the 'DigitalAssets'
              table
input : value
output : INSERT INTO DigitalAssets (name, symbol, total_supply) VALUES ('CryptoPet', 'CPT', 1000000000);
correct_answer : INSERT INTO DigitalAssets (name, symbol, total_supply) VALUES ('CryptoPet', 'CPT', 1000000000);
```
```
instruction : What is the average age of patients who received therapy from counselors named \"James\" or \"Michelle\"?
input : value
output : SELECT AVG(patients.age) FROM patients JOIN counselors ON patients.counselor_id = counselors.counselor_id WHERE counselors.name IN
              ('James', 'Michelle');
correct_answer : SELECT AVG(patients.age) FROM patients JOIN counselors ON patients.counselor_id = counselors.counselor_id WHERE
              counselors.name IN ('James', 'Michelle');
```
```
instruction : Insert new records for artists from underrepresented communities.
input : value
output : INSERT INTO Artists (id, name, community) VALUES (1, 'Jane Doe', 'Underrepresented Community'), (2, 'Jim Smith', 'Underrepresented
              Community');
answer : INSERT INTO Artists (id, name, community) VALUES (1, 'Jane Doe', 'Underrepresented Community'), (2, 'Jim Smith', 'Underrepresented
              Community');
```
```
instruction : Insert a new record into the 'authors' table with the name 'Alex Brown' and newspaper 'The Washington Post'
input : value
output : INSERT INTO authors (name, newspaper) VALUES ('Alex Brown', 'The Washington Post');
correct_answer : INSERT INTO authors (name, newspaper) VALUES ('Alex Brown', 'The Washington Post');
```

Table 4: Examples of Syntatic-Text-to-SQL Records. Each query sample consists of an "instruction" concatenated with an "input" (which is always an empty string for this dataset), while the "answer" represents the ground-truth label of the dataset. The "output" is a demonstration of the LLM's response to the query sample.

## A.2 Model Details

We consider the following representative LLMs in our empirical evaluation across different architectures, parameter counts, and design philosophies: **T5-Large** [3], **LLaMA-7B** [22], **OPT-6.7B** [23], **BLOOM-7B** [24], and **GPT-J-6B** [25]. T5-Large employs an encoder-decoder transformer model, processing input text through an encoder and generating output text via a decoder, making it particularly suitable for text-to-text tasks. In contrast, LLaMA-7B, OPT-6.7B, BLOOM-7B, and GPT-J-6B utilize decoder-only architectures optimized for autoregressive text generation. These models have parameter counts ranging from 770 million (T5-Large) to over 7 billion (BLOOM-7B), covering a standard and reasonable range for empirical investigation in scientific research. The design philosophies also vary significantly: T5-Large focuses on converting all tasks into a text-to-text format, while BLOOM-7B emphasizes multilingual capabilities, supporting 59 languages and 12 programming languages. LLaMA-7B and GPT-J-6B prioritize openness and efficiency, aiming to enhance accessibility and performance in NLP, while OPT-6.7B targets transparency and competitive performance.

The hyper-parameters during fine-tuning are listed in Table 5.

---

[5]`https://huggingface.co/google-t5/t5-large`
[6]`https://huggingface.co/yahma/llama-7b-hf`
[7]`https://huggingface.co/facebook/opt-6.7b`
[8]`https://huggingface.co/bigscience/bloom-7b1`
[9]`https://huggingface.co/EleutherAI/gpt-j-6b`

| | T5-Large | Llama-7B | OPT-6.7B | BLOOM-7B | GPT-J-6B |
|---|---|---|---|---|---|
| Parameters | 770M | 6.7B | 6.7B | 7.1B | 6.1B |
| Learning Rate | 1e-3 | 3e-4 | 1e-3 | 3e-4 | 2e-3 |
| Batch Size | 128 | 32 | 32 | 32 | 32 |
| Micro Batch Size | 32 | 8 | 8 | 8 | 8 |
| Maximum Length | 512 | 256 | 256 | 256 | 256 |
| Model Source | 5 | 6 | 7 | 8 | 9 |

Table 5: Hyper-parameters of LLMs during fine-tuning.

## A.3 LLM Adaptation

By default, each LLM is fine-tuned for 5 epochs. For **LoRA**, we set the rank to 8 and the alpha value to 16, and tune the attention vectors $q$, $k$, and $v$. For **Top2Head-tuning**, only the first 2 top layers are tuned. In **Adapter-H**, we add an intermediate projection layer with size 256 and apply "tanh" as the nonlinear activation function. For **Prefix-tuning**, the number of virtual tokens is set to 30. In **P-tuning**, the encoder size is set to 128, with 20 virtual tokens. For **Prompt-tuning**, the initial prompt is chosen to be "Complete the following task: ".

## A.4 Attack Implementation

For the **Likelihood-ref** attack, following the original implementation [12], we use the original pre-trained model (which was not adapted using the domain data) as the reference model. For the **Neighborhood** attack, we set the size of the neighbor candidates to 25 and the word mask rate to 0.3. Additionally, aligned with the original paper [29], we use a third-party BERT model[10] from Huggingface to generate the neighbors of a given query sample. For **Min-K** and **Min-K++**, we set $K$ to 0.2, and both the window size and stride with respect to N-gram to 1.

Evaluating the attack AUC-ROC involves measuring the entire area under the ROC curve, which corresponds to varying thresholds $\tau$ of the membership score. In contrast, measuring the attack FPR@0.1% TPR or FPR@1% TPR involves selecting the threshold $\tau$ to match a specific true positive rate (0.1% or 1%) on the query set and then evaluating the corresponding false positive rates.

## B Additional Results

We present the overall quantitative results of evaluating different attack methods across various metrics and LLMs fine-tuned with LoRA on different datasets in Tables 6-8. These results supplement the findings illustrated in Figure 3 of the main paper.

We present in Tables 9-11 the quantitative results of the utility (measured by model accuracy) and attack performance (evaluated with AUC-ROC) when comparing different adaptation methods across different data sizes (Table 9), fine-tuning epochs (Table 10), and model sizes (Table 11) on the CorpClimate dataset. We use by default the OPT-6.7B model as the target LLM. These results are supplementary to Figures 5 & 6 in the main paper.

---

[10]https://huggingface.co/google-bert/bert-base-multilingual-cased

| Attack Method | Metric | Model | | | | |
|---|---|---|---|---|---|---|
| | | T5-Large | Llama-7B | OPT-6.7B | BLOOM-7B | GPT-J-6B |
| Likelihood | AUC-ROC | 0.54 | 0.52 | 0.52 | 0.51 | 0.54 |
| | FPR(%)@0.1%TPR | 0.71 | 0.00 | 0.00 | 0.20 | 0.17 |
| | FPR(%)@1%TPR | 2.33 | 1.63 | 0.00 | 0.89 | 1.06 |
| Likelihood-ref | AUC-ROC | 0.62 | 0.62 | 0.60 | 0.57 | 0.59 |
| | FPR(%)@0.1%TPR | 5.83 | 5.62 | 5.47 | 4.92 | 4.68 |
| | FPR(%)@1%TPR | 12.08 | 11.73 | 9.86 | 8.77 | 9.03 |
| Zlib Entropy | AUC-ROC | 0.53 | 0.54 | 0.52 | 0.54 | 0.51 |
| | FPR(%)@0.1%TPR | 0.31 | 0.00 | 0.00 | 0.29 | 0.00 |
| | FPR(%)@1%TPR | 1.03 | 2.22 | 1.00 | 1.88 | 0.74 |
| Neighborhood | AUC-ROC | 0.52 | 0.53 | 0.53 | 0.52 | 0.52 |
| | FPR(%)@0.1%TPR | 0.00 | 0.00 | 0.02 | 0.00 | 0.01 |
| | FPR(%)@1%TPR | 0.00 | 0.00 | 1.05 | 0.22 | 0.69 |
| Min-K | AUC-ROC | 0.52 | 0.52 | 0.52 | 0.53 | 0.52 |
| | FPR(%)@0.1%TPR | 0.00 | 0.38 | 0.00 | 0.00 | 0.00 |
| | FPR(%)@1%TPR | 0.00 | 1.17 | 0.00 | 0.24 | 0.00 |
| Min-K++ | AUC-ROC | 0.53 | 0.52 | 0.52 | 0.54 | 0.52 |
| | FPR(%)@0.1%TPR | 0.00 | 0.38 | 0.00 | 0.00 | 0.00 |
| | FPR(%)@1%TPR | 0.00 | 1.17 | 0.00 | 0.24 | 0.00 |
| Gradient-Norm | AUC-ROC | 0.63 | 0.60 | 0.58 | 0.54 | 0.55 |
| | FPR(%)@0.1%TPR | 3.49 | 3.31 | 4.57 | 3.13 | 3.22 |
| | FPR(%)@1%TPR | 8.87 | 9.93 | 11.28 | 8.49 | 7.98 |

Table 6: Overall attack effectiveness across different LLMs fine-tuned with LoRA (SQL).

| Attack Method | Metric | Model | | | | |
|---|---|---|---|---|---|---|
| | | T5-Large | Llama-7B | OPT-6.7B | BLOOM-7B | GPT-J-6B |
| Likelihood | AUC-ROC | 0.63 | 0.61 | 0.61 | 0.58 | 0.56 |
| | FPR(%)@0.1%TPR | 1.89 | 2.32 | 2.17 | 0.70 | 1.28 |
| | FPR(%)@1%TPR | 10.08 | 11.12 | 13.67 | 5.92 | 6.01 |
| Likelihood-ref | AUC-ROC | 0.70 | 0.71 | 0.73 | 0.71 | 0.70 |
| | FPR(%)@0.1%TPR | 5.85 | 6.43 | 5.87 | 3.08 | 3.25 |
| | FPR(%)@1%TPR | 16.62 | 21.11 | 15.44 | 13.31 | 12.99 |
| Zlib Entropy | AUC-ROC | 0.62 | 0.62 | 0.63 | 0.66 | 0.63 |
| | FPR(%)@0.1%TPR | 1.85 | 4.56 | 3.17 | 2.98 | 4.14 |
| | FPR(%)@1%TPR | 7.73 | 14.64 | 10.05 | 8.85 | 12.21 |
| Neighborhood | AUC-ROC | 0.67 | 0.64 | 0.62 | 0.63 | 0.65 |
| | FPR(%)@0.1%TPR | 1.81 | 2.33 | 2.18 | 1.59 | 5.54 |
| | FPR(%)@1%TPR | 5.42 | 9.96 | 8.87 | 10.07 | 11.12 |
| Min-K | AUC-ROC | 0.50 | 0.58 | 0.56 | 0.58 | 0.53 |
| | FPR(%)@0.1%TPR | 0.00 | 1.64 | 0.81 | 0.68 | 0.00 |
| | FPR(%)@1%TPR | 0.00 | 7.90 | 1.82 | 2.79 | 0.00 |
| Min-K++ | AUC-ROC | 0.51 | 0.58 | 0.56 | 0.57 | 0.54 |
| | FPR(%)@0.1%TPR | 0.00 | 2.04 | 1.01 | 0.73 | 0.00 |
| | FPR(%)@1%TPR | 0.00 | 6.54 | 3.99 | 4.24 | 0.00 |
| Gradient-Norm | AUC-ROC | 0.73 | 0.71 | 0.72 | 0.71 | 0.71 |
| | FPR(%)@0.1%TPR | 5.73 | 6.22 | 5.86 | 5.99 | 4.83 |
| | FPR(%)@1%TPR | 14.98 | 18.69 | 17.41 | 18.16 | 15.52 |

Table 7: Overall attack effectiveness across different LLMs fine-tuned with LoRA (Sujet Finance).

| Attack Method | Metric | Model | | | | |
|---|---|---|---|---|---|---|
| | | T5-Large | Llama-7B | OPT-6.7B | BLOOM-7B | GPT-J-6B |
| Likelihood-based | AUC-ROC | 0.59 | 0.58 | 0.57 | 0.58 | 0.61 |
| | FPR(%)@0.1%TPR | 1.19 | 1.41 | 1.08 | 1.08 | 2.87 |
| | FPR(%)@1%TPR | 9.08 | 5.69 | 4.99 | 5.19 | 8.83 |
| Zlib Entropy-based | AUC-ROC | 0.65 | 0.63 | 0.62 | 0.56 | 0.63 |
| | FPR(%)@0.1%TPR | 2.59 | 3.18 | 2.02 | 0.63 | 1.16 |
| | FPR(%)@1%TPR | 10.07 | 9.89 | 8.88 | 3.94 | 9.46 |
| Neighborhood | AUC-ROC | 0.62 | 0.65 | 0.61 | 0.63 | 0.65 |
| | FPR(%)@0.1%TPR | 1.64 | 3.13 | 1.11 | 1.26 | 2.89 |
| | FPR(%)@1%TPR | 6.07 | 7.25 | 6.01 | 6.35 | 7.77 |
| Min-K-based | AUC-ROC | 0.57 | 0.61 | 0.59 | 0.63 | 0.62 |
| | FPR(%)@0.1%TPR | 1.02 | 2.08 | 2.21 | 2.53 | 3.03 |
| | FPR(%)@1%TPR | 2.13 | 5.19 | 6.21 | 7.77 | 8.12 |
| Min-K++-based | AUC-ROC | 0.59 | 0.62 | 0.65 | 0.65 | 0.66 |
| | FPR(%)@0.1%TPR | 2.15 | 2.61 | 2.97 | 3.33 | 3.59 |
| | FPR(%)@1%TPR | 3.34 | 5.92 | 6.48 | 8.09 | 8.15 |
| Refernce-based | AUC-ROC | 0.72 | 0.74 | 0.75 | 0.72 | 0.70 |
| | FPR(%)@0.1%TPR | 6.79 | 7.82 | 7.19 | 6.48 | 6.14 |
| | FPR(%)@1%TPR | 15.03 | 19.88 | 18.75 | 16.87 | 15.33 |
| Gradient-Norm-based | AUC-ROC | 0.72 | 0.73 | 0.71 | 0.72 | 0.72 |
| | FPR(%)@0.1%TPR | 6.79 | 6.94 | 6.48 | 6.82 | 7.05 |
| | FPR(%)@1%TPR | 14.09 | 17.18 | 18.44 | 15.02 | 16.63 |

Table 8: Overall attack effectiveness across different LLMs fine-tuned with LoRA (CorpClimate).

| Metric | Data Size | Adaptation Technique | | | | | | |
|---|---|---|---|---|---|---|---|---|
| | | Full | Prefix-tuning | Top2-Head | LoRA | P-tuning | Adapter-H | Prompt-tuning |
| Model Accuracy | 25%(2790) | 0.424 | 0.549 | 0.523 | 0.526 | 0.519 | 0.531 | 0.479 |
| | 50%(5580) | 0.521 | 0.558 | 0.541 | 0.579 | 0.542 | 0.563 | 0.562 |
| | 75%(8370) | 0.653 | 0.657 | 0.652 | 0.654 | 0.647 | 0.655 | 0.652 |
| | full(11159) | 0.674 | 0.669 | 0.668 | 0.671 | 0.665 | 0.671 | 0.666 |
| Attack AUC | 25%(2790) | 0.794 | 0.769 | 0.761 | 0.76 | 0.758 | 0.749 | 0.745 |
| | 50%(5580) | 0.759 | 0.759 | 0.755 | 0.752 | 0.749 | 0.748 | 0.741 |
| | 75%(8370) | 0.757 | 0.755 | 0.753 | 0.749 | 0.746 | 0.742 | 0.731 |
| | full(11159) | 0.751 | 0.751 | 0.749 | 0.747 | 0.737 | 0.737 | 0.729 |

Table 9: Comparison of various adaptation techniques across different fine-tuning dataset sizes (CorpClimate) on the OPT-6.7B model. The attack AUC-ROC is evaluated using the Likelihood-ref approach. The shaded column indicates the varying dataset sizes (ranging from 25% to the full dataset) used for adapting the model, with the absolute number of samples presented in brackets.

| Metric | Epoch | Adaptation Technique | | | | | | |
|---|---|---|---|---|---|---|---|---|
| | | Full | Prefix-tuning | Top2-Head | LoRA | P-tuning | Adapter-H | Prompt-tuning |
| Model Accuracy | 1 | 0.404 | 0.436 | 0.444 | 0.595 | 0.503 | 0.447 | 0.442 |
| | 2 | 0.508 | 0.528 | 0.525 | 0.653 | 0.588 | 0.556 | 0.547 |
| | 3 | 0.597 | 0.577 | 0.597 | 0.664 | 0.642 | 0.596 | 0.617 |
| | 4 | 0.668 | 0.622 | 0.656 | 0.669 | 0.657 | 0.651 | 0.661 |
| | 5 | 0.673 | 0.669 | 0.673 | 0.671 | 0.664 | 0.669 | 0.669 |
| Attack AUC | 1 | 0.679 | 0.651 | 0.649 | 0.644 | 0.641 | 0.638 | 0.633 |
| | 2 | 0.709 | 0.698 | 0.688 | 0.685 | 0.673 | 0.67 | 0.655 |
| | 3 | 0.748 | 0.744 | 0.739 | 0.724 | 0.711 | 0.707 | 0.696 |
| | 4 | 0.753 | 0.751 | 0.746 | 0.739 | 0.737 | 0.737 | 0.735 |
| | 5 | 0.755 | 0.753 | 0.752 | 0.747 | 0.745 | 0.742 | 0.741 |

Table 10: Comparison of different adaptation techniques across various fine-tuning epochs (CorpClimate) on the OPT-6.7B model. The attack AUC-ROC is evaluated using the Likelihood-ref approach. The shaded column indicates the varying fine-tuning epochs (ranging from 1 to the default value of 5) used for adapting the model.

| Metric | Model (Size) | Adaptation Technique | | | | | | |
|---|---|---|---|---|---|---|---|---|
| | | Full | Prefix-tuning | Top2-Head | LoRA | P-tuning | Adapter-H | Prompt-tuning |
| Model Accuracy | OPT-125M | 0.669 | 0.663 | 0.651 | 0.661 | 0.645 | 0.652 | 0.641 |
| | OPT-350M | 0.673 | 0.671 | 0.653 | 0.661 | 0.656 | 0.661 | 0.643 |
| | OPT-1.3B | 0.673 | 0.675 | 0.662 | 0.663 | 0.666 | 0.665 | 0.649 |
| | OPT-2.7B | 0.678 | 0.681 | 0.665 | 0.667 | 0.667 | 0.671 | 0.653 |
| | OPT-6.7B | 0.685 | 0.681 | 0.669 | 0.671 | 0.673 | 0.675 | 0.672 |
| Attack AUC-ROC | OPT-125M | 0.699 | 0.693 | 0.689 | 0.683 | 0.681 | 0.677 | 0.668 |
| | OPT-350M | 0.714 | 0.704 | 0.691 | 0.688 | 0.685 | 0.681 | 0.668 |
| | OPT-1.3B | 0.721 | 0.713 | 0.709 | 0.694 | 0.689 | 0.688 | 0.679 |
| | OPT-2.7B | 0.727 | 0.719 | 0.717 | 0.711 | 0.702 | 0.694 | 0.688 |
| | OPT-6.7B | 0.767 | 0.751 | 0.749 | 0.747 | 0.741 | 0.738 | 0.735 |

Table 11: Comparison of different adaptation techniques across various model sizes (CorpClimate). The attack AUC-ROC is evaluated using the Likelihood-ref approach. The shaded column indicates the varying target model size (ranging from 125M to the default value of 6.7B).

