# OpenReview forum: "PrivAuditor: Benchmarking Data Protection Vulnerabilities in LLM Adaptation Techniques"
_NeurIPS.cc/2024/Datasets_and_Benchmarks_Track — NeurIPS 2024 Track Datasets and Benchmarks Spotlight_

### Official Review · Reviewer_6BzH · 2024-06-21
**Comprehensive, solid, and insightful assessment of membership inference risks introduced through LLM-finetuning**

**Rating:** 8
**Confidence:** 5
**Correctness:** yes
**Clarity:** yes, absolutely

**Review:**

The paper is remarkably well-written, -readable and -organized. Figures are reasonable, insightful, and also well-readable and the line of argument is always clear and easy to follow.

Content-wise, the subject of the paper is highly relevant and timely and the considerations underlying it are highly reasonable. The experiment design, the selection of LLMs, adaptation datasets, attack approaches, etc., and the experiment execution itself are always solid and sound. The number of experiments executed is impressive and allows for highly valuable insights of remarkable practical relevance for any player adapting a pre-trained LLM to usecase-specific needs with not-to-be-revealed domain data.

Leaving aside absolutely minor and easily fixable issues (see below), I see this paper as a strong, insightful, and highly valuable contribution to the datasets and benchmarks track.

**Strengths:**

- highly relevant and timely subject
- absolutely solid design and execution of experiments
- impressive number of structured experiments across LLMs, attacks, adaptation techniques, adaptation datasets, ... with significant and highly valuable insights stemming from experiment results
- Scientifically rigid from the beginning to the end

**Additional Feedback:**

n/a

**Documentation:**

yes, all necessary details there

**Ethics:**

no concerns

**Limitations:**

limitations should (and could easily) be added. Negative societal impart probably not applicable.

**Opportunities For Improvement:**

Only some minor things:
- Research questions could possibly be presented earlier in the paper
- I would have enjoyed to read some implications and at least a brief discussion of limitations towards the end of the paper
- RW could be a bit more exhaustive and explanatory / contextualizing
- in case this pushes towards page limits, consider streamlining 2.2 and, in particular, think about whether all formalizations are actually needed (also for really well-established measures/attacks with respective papers being cited anyhow)

Besides, some minor-minor things:
- in figure 3, the grey "Neighboor" seems to be an artifac - "no such bar in barchart"
- in figure 3, the three sub-diagrams should all be scaled from 0.50 to 0.75 y-wise. Having the first with 0.5 to 0.64 is not helpful, especially given that the SQL case is compared to the others in the third para of 5.3
- maybe switch figures 5 and 6, given that 6 is currently mentioned earlier than 5
- first para of 5.5: referenced figure not existing

**Relation To Prior Work:**

see above: RW is fine but could be a bit more exhaustive and explanatory / contextualizing

**Summary And Contributions:**

This paper presents an experimental assessment of the privacy risks introduced during the finetuning of state-of-the-art LLMs. In particular, it considers the widely recognized membership inference risks, which is highly relevant in the context of custom-adapted LLMs: How high is the risk of being able to identify certain tokens being included in the dataset used for finetuning/customizing a based model? If such identification is possible, attackers may be able to learn confidential information about said dataset, which LLM operators may want /have to keep confidential for legal or business reasons.  Being aware of this risk and the factors influencing it is thus of outstanding practical relevance.

To assess these inference risks, the authors employ different likelihood metrics (simple likelihood, neighborhood-based, etc) and apply them to the main re-trainable LLM applying the most important and established adaptation techniques to each of these for three different adaptation datasets. Based on the resulting remarkable number of experiments, they assess whether the data used for pre-training can be leaked (or, more precisely, how high the leakage risk is , according to said metrics), how the different available adaptation techniques influence the risk, and which further factors affect it.

Experiment results show, among other insights, that the employed adaptation technique has a significant impact on the resulting inference risk, that larger models are more subject to this risk than smaller models, or that and how the different risk metrics "behave" differnetly across the different LLMs. Besides the experimental results, another contribution of the paper is the general approach for experimentally assessing the membership inference risk arising in the context of LLM adaptation and the respective experiment method including the selected retraining datasets.

---

> ### Author Rebuttal · Authors · 2024-08-16
>
> We sincerely thank the reviewers for their constructive feedback! We are pleased that most reviewers have a clear positive view of our submission, praising it as a **"strong, insightful, and highly valuable contribution"**, **"remarkably well-written, readable, and organized"**, **"entirely self-contained, understandable, and even intuitive to the reader"**, and highlighting the **"absolutely solid design and execution of experiments"**.
>
>
> We below address the individual concerns of **Reviewer  6BzH**. All minor points will be directly incorporated in the revised manuscript.
>
>
>
> **[1.Research questions could possibly be presented earlier in the paper]**
>
> While we structured the current version to provide the necessary background and context before introducing the research questions, we will certainly consider your feedback and explore ways to bring them forward if feasible within the flow of the paper.
>
> **[2. I would have enjoyed to read some implications and at least a brief discussion of limitations towards the end of the paper]**
>
> We greatly appreciate the suggestion and will include a limitations section discussing potential extensions of our work, such as examining assumptions in common threat models, incorporating future attack types, and considering diverse data modalities with varying characteristics.
>
> **[3. RW could be a bit more exhaustive and explanatory / contextualizing]**
>
> We thank the suggestion and will make the related work section more exhaustive by providing additional explanations and context to better frame our contributions within the existing literature.
>
> **[4. Minor points: Figure 3, order of Figure 5/6 , reference to Figure 5]**
>
> We sincerely thank the reviewer for pointing these out and have already updated our manuscript accordingly.

---

### Official Review · Reviewer_5k61 · 2024-07-03
**First benchmark for privacy leakage risk evaluation in LLMs.**

**Rating:** 10
**Confidence:** 5
**Clarity:** Paper and results are presented and w…

**Review:**

The paper is of high quality. The concept and the explanation of such a difficult topic in a clear way is consistent throughout the paper. This work is significant and relevant to these times where there are a number of LLMs out there but they pose a lot of security concerns. Privacy vulnerability is one of the serious concerns.

Benchmarking the methods to understand the privacy concerns in LLMs is a significant step and this paper clearly explains the approach. This study examines the training data membership leakage risk in mainstream large language models based on encoder-decoder and decoder only structures. Benchmarking results highlight the privacy leakage risks during LLM adaptation and further propose a unified platform to measure these potential privacy risks.

The benchmarking results show that privacy vulnerabilities are influenced due to the size of domain data applied for training, number of fine-tuning iterations, size of trainable parameters.

**Strengths:**

The submission presents the benchmark to investigate the privacy implications in LLMs adaptation techniques and their comprehensive study.

Privacy concerns in LLMs is a well known issue. LLMs are known to spit out the personal or private details due to the leakage of training data membership. This benchmark evaluates a number of LLMs for privacy concerns and highlights the importance of developing privacy preserving adaptation techniques in practice.

The benchmark shows the privacy  vulnerabilities are influenced due to the size of domain data applied for training, number of fine-tuning iterations, size of trainable parameters.

**Additional Feedback:**

1. On page 8 in section 5.5: Put the correct figure number which shows a question mark at the moment.

2. Discuss the limitations of current work and in what direction the existing work could be extended.

**Correctness:**

The details in the submission are described in a sound way. The evaluation methods and experiment design is appropriate.

**Documentation:**

This benchmark provides sufficient details to support reproducibility and the url for the code and data access.

**Ethics:**

No concerns.

**Limitations:**

1. Authors examine the training data membership leakage risk in mainstream LLMs based on encoder-decoder and decoder-only structures. However, they did not discuss the limitations of these structures and what would be the difference If they consider other structures.

2. I would suggest discussing the limitations of their current work and in what direction the existing work could be extended.

**Opportunities For Improvement:**

1. Authors examine the training data membership leakage risk in mainstream LLMs based on encoder-decoder and decoder-only structures. However, they did not discuss the limitations of these structures and what would be the difference If they consider other structures.

2. I would suggest discussing the limitations of their current work and in what direction the existing work could be extended.

**Relation To Prior Work:**

This benchmark does not explicitly discuss how this work differs from previous contributions.
This is acceptable for this manuscript since this presents first benchmark results.

**Summary And Contributions:**

This paper presents the first benchmark investigating the privacy leakage risks during adaptation techniques in LLMs, accompanied by a comprehensive empirical study. The privacy vulnerabilities of LLMs are evaluated through the lens of membership inference attacks (MIA) due to the extensive applicability of MIAs. Different attack approaches, Likelihood-based, Likelihood with Reference, Zlib entropy as reference, neighborhood based, min-K% probability, Min-K%++, Gradient Norm based and adaptation techniques (full fine tuning, top-2 head tuning, adapter-H, Prefix tuning, LoRA, P-tuning, Prompt tuning and in-context learning) are evaluated.

Authors examine the training data membership leakage risk in mainstream LLMs based on encoder-decoder and decoder only structures. The paper highlights the privacy leakage risks during LLM adaptation and proposes a unified platform to measure these potential privacy risks.

---

> ### Author Rebuttal · Authors · 2024-08-16
>
> We sincerely thank the reviewers for their constructive feedback! We are pleased that most reviewers have a clear positive view of our submission, praising it as a **"strong, insightful, and highly valuable contribution"**, **"remarkably well-written, readable, and organized"**, **"entirely self-contained, understandable, and even intuitive to the reader"**, and highlighting the **"absolutely solid design and execution of experiments"**.
>
> We below address the individual concerns of **Reviewer  5k61**. All minor points will be directly incorporated in the revised manuscript.
>
> **[1. Authors examine the training data membership leakage risk in mainstream LLMs based on encoder-decoder and decoder-only structures. However, they did not discuss the limitations of these structures and what would be the difference If they consider other structures.]**
>
> We thank the reviewer for these suggestions. While it is challenging, if not impossible, to make general claims about the privacy leakage characteristics of different model structures, we do observe differences in the potential attack surface depending on the common application scenarios associated with each model type. We would be glad to incorporate this discussion in the revision.
>
> **[2. I would suggest discussing the limitations of their current work and in what direction the existing work could be extended.]**
>
> We greatly appreciate the suggestion and will include a limitations section discussing potential extensions of our work, such as examining assumptions in common threat models, incorporating future attack types, and considering diverse data modalities with varying characteristics.

---

> > ### Comment · Reviewer_5k61 · 2024-08-22
> >
> > Look forward to the revised version of paper with suggested updates.

---

### Official Review · Reviewer_owSp · 2024-07-18
**A well-written, well-executed benchmark of privacy auditing for LLM adaptation**

**Rating:** 8
**Confidence:** 4
**Correctness:** Yes.
**Clarity:** Very. See **Strengths** above.

**Review:**

I will co-locate strengths and weaknesses here, and use this for most of my review, in order to avoid repeating myself. At a high level, I think this is a well-written, well-executed paper that should be accepted.

## Strengths

I cannot praise the clarity of writing enough in this work. The authors waste no time on superfluous discussion, and instead devote a significant amount of effort and writing into being as clear and direct as possible. Section 2 is incredibly effective at giving a reader a brief, but useful survey of privacy auditing metrics, and does a terrific job in both (1) putting these metrics in context with prior work, and (2) showing how many of them can be viewed as thematically similar, with important things like normalization factors that differentiate them (and their utility). Similar strength of writing occurs in Section 3, and really throughout.

I think the benchmark design (Section 5.2) is worth calling out specifically. The authors pose three general questions (which are at this point, due to the clarity of writing, entirely self-contained, understandable, and even intuitive to the reader), and then design perhaps as straightforward a set of experiments to answer that question as possible. It's really useful seeing how the experiments were designed for each research question separately, and I really enjoyed reading it. This benefits the reader in Section 5.3, as it focuses the discussion of results in a way that aids understanding.

In short, the benchmark is well-designed, well-explained, and interesting enough to warrant acceptance.

## Weaknesses

I think the biggest weakness of the paper is in a lack of details about dataset overlap. The dataset overlap between pre-training and fine-tuning is extremely important in these types of evaluations. The authors simply claim that their datasets "do not appear in the pre-training datasets of the studied foundation models," but give no evidence to back this up.

It's worth noting that there are really 2 things that could be shown:

1. Nominally, the adaptation datasets are not sub-components of the pre-training datasets. This likely can be demonstrated by detailing what the pre-training datasets are (or how they were collected, which I believe is stated for at least some of the models trained).
1. That there is no overlap in examples between pre-training and adaptation. This is harder. Web-scraped corpora may have random examples that do appear in these datasets. I don't know if there's an easy answer here, but it may be worth showcasing membership inference attacks on pre-trained models **without adaptation**.

Another weakness of the paper is that the issue of "how severe is this privacy leakage" is not really addressed. Consider Section 5.3. In the first paragraph, the authors state that "the results reveal clear distinctions in the distributions" in Figure 2. But relative to what? It is not clear to me that they actually represent significant privacy leakage. Moreover, the actual AUC-ROC values (Table 5) seem to be close to 0.5, which essentially means they are no better than random guessing. To that end, I wish the authors were a bit more nuanced in their discussion here. Similarly the discussion in Section 5.3. It claims that "[Membership Inference Attacks Detect Data Used for LLM Adaptation." Again, I think there is reason to be nuanced in the discussion. The fact that some methods are essentially doing no better than random is important.

### Minor weaknesses

* I couldn't see any results using in-context learning. They don't seem to be in Figure 6 even though it is mentioned in the set of adaptation techniques used.
* Undefined Figure in Section 5.5.
* The paper doesn't seem to have used the datasets & benchmarks template exactly (e.g. there are no line numbers).

**Strengths:**

See **Strengths** above.

**Additional Feedback:**

None.

**Documentation:**

The documentation seems generally correct. I will note that the documentation also does not seem to suggest that in-context learning is a part of the benchmark.

**Ethics:**

None.

**Limitations:**

I didn't really see any discussion of this. I would like to see the authors acknowledge the limitations of membership inference attacks in LLM contexts, as evidenced by the fact that some MIAs are no better than random on some datasets.

**Opportunities For Improvement:**

See **Weaknesses** above.

**Relation To Prior Work:**

Generally yes. I think the one paper that might bear some discussion that is missing is [Duan et al., "Do Membership Inference Attacks Work on Large Language Models?"], and the demonstration that (at least in pre-training) LLM settings, MIAs can often not perform well.

**Summary And Contributions:**

This paper benchmarks a variety of privacy auditing techniques on a variety of "adapted" pre-trained large language models. These models are adapted to specific datasets using a variety of techniques (e.g. LoRA, and full fine-tuning). The authors of the benchmark then pose 3 specific research questions around privacy, adaptation, and LLMs, and design a set of experiments to help address each one. Finally, the authors give the results of these experiments, and discuss their findings.

---

> ### Author Rebuttal · Authors · 2024-08-16
>
> We sincerely thank the reviewers for their constructive feedback! We are pleased that most reviewers have a clear positive view of our submission, praising it as a **"strong, insightful, and highly valuable contribution"**, **"remarkably well-written, readable, and organized"**, **"entirely self-contained, understandable, and even intuitive to the reader"**, and highlighting the **"absolutely solid design and execution of experiments"**.
>
> We below address the individual concerns of **Reviewer  owSp**. All minor points will be directly incorporated in the revised manuscript.
>
> **[1. I think the biggest weakness of the paper is in a lack of details about dataset overlap. The dataset overlap between pre-training and fine-tuning is extremely important in these types of evaluations. The authors simply claim that their datasets "do not appear in the pre-training datasets of the studied foundation models," but give no evidence to back this up.]**
>
> We appreciate the reviewer’s suggestion and will include more details about the dataset overlap in the revision. We indeed have taken certain measures to minimize the potential overlap between the fine-tuning datasets and the pre-training dataset:
> - The selected fine-tuning datasets were all released *after* the pre-trained models were created
> - Extensive pre-processing has been conducted before the fine-tuning datasets were compiled, making it highly unlikely that any data points overlap with the pre-training dataset, even if (with a very low probability) they may originate from the same source.
> - We also included synthetic data with a specific structure that does not derive from web data
>
> **[2.Another weakness of the paper is that the issue of "how severe is this privacy leakage" is not really addressed. Consider Section 5.3. In the first paragraph, the authors state that "the results reveal clear distinctions in the distributions" in Figure 2. But relative to what? It is not clear to me that they actually represent significant privacy leakage. Moreover, the actual AUC-ROC values (Table 5) seem to be close to 0.5, which essentially means they are no better than random guessing. To that end, I wish the authors were a bit more nuanced in their discussion here. Similarly the discussion in Section 5.3. It claims that "[Membership Inference Attacks Detect Data Used for LLM Adaptation." Again, I think there is reason to be nuanced in the discussion. The fact that some methods are essentially doing no better than random is important.]**
>
> We will expand the discussion on this point. Our assessment of the privacy risk is based primarily on the most successful attack results, as all practically implementable attacks represent only a lower bound when estimating potential risk. We recognize that some methods may be less effective in certain scenarios and will emphasize this more clearly in the revision.
>
> **[3. I couldn't see any results using in-context learning. They don't seem to be in Figure 6 even though it is mentioned in the set of adaptation techniques used.]**
>
> We thank the reviewer for pointing this out and will revise the manuscript accordingly. The in-context learning results are included in the overall quantitative results presented in Table 8 in the Appendix. However, we deliberately excluded in-context learning from the qualitative comparisons shown in Figures 5 and 6, as its non-parameter-update nature makes the application and evaluation scenarios not directly comparable to the others.

---

### Official Review · Reviewer_fLBd · 2024-07-25
**Benchmarking Data Protection Vulnerabilities in LLM Adaptation Techniques**

**Rating:** 6
**Confidence:** 4
**Correctness:** Yes
**Clarity:** Yes

**Review:**

Pros:
This paper made a good attempt to explore how different member inference attack strategies and different adaptation of LLMs affect the privacy leakage conditions. It’s quite useful to know more about the privacy risk for Lora or other efficient fine-tuning methods. This paper also try to include comprehensive studies about different factors, including data size, model size, and number of trainable parameters.

Cons:

1. The most unique and interesting factor of the finetuning process is the trainable parameter size. While the conclusion seems not very consistent. The result in Figure 6 does not align with the conclusion made in section 5.4. Models with more trainable parameters don’t always show more privacy leakage, especially at early epochs and smaller data sizes.

2. It shows in Figure 3 that except for Likelihood-ref and Gradient Norm, other member inference attacks couldn’t achieve relatively high performance. It would be good to see some explanations about the difference.

**Strengths:**

It’s been missing in the existing literature about the relationship between privacy leakage and different fine-tuning strategies. It’s well-motivated and a very practical perspective to study the privacy problem for efficient fine-tuning and in-context learning.

**Additional Feedback:**

No

**Documentation:**

Yes

**Limitations:**

Yes

**Opportunities For Improvement:**

According to the comments in the review section, It would be interesting to see more experiments about the relationship between trainable parameters and member inference performance.

**Relation To Prior Work:**

Yes

**Summary And Contributions:**

This benchmark mainly focuses on exploring how different adaption methods influence the privacy risk of LLMs. This paper conducts throughout experiments on different member inference attack methods and different LLM adaptations. Through experiments, this paper draw conclusions on to what extent finetuning data could be detected by member inference attacks and what types of factors would affect the inference performance.

---

> ### Author Rebuttal · Authors · 2024-08-16
>
> We sincerely thank the reviewers for their constructive feedback! We are pleased that most reviewers have a clear positive view of our submission, praising it as a **"strong, insightful, and highly valuable contribution"**, **"remarkably well-written, readable, and organized"**, **"entirely self-contained, understandable, and even intuitive to the reader"**, and highlighting the **"absolutely solid design and execution of experiments"**.
>
> We below address the individual concerns of **Reviewer  fLBd**. All minor points will be directly incorporated in the revised manuscript.
>
> **[1. The most unique and interesting factor of the finetuning process is the trainable parameter size. While the conclusion seems not very consistent. The result in Figure 6 does not align with the conclusion made in section 5.4. Models with more trainable parameters don’t always show more privacy leakage, especially at early epochs and smaller data sizes.]**
>
> We are thankful for the valuable feedback and would like to clarify that our conclusion is indeed consistent with the experimental results. We will revise the manuscript to prevent potential confusion. The relevant experimental results concerning the trainable parameter size are primarily displayed in the right-most subfigure of Figure 5 (showing attack performance) and Figure 6 (showing utility). The trend of privacy leakage can be observed by:
> - Examining each curve in the right-most subfigure of Figure 5 from left to right, as larger models with more trainable parameters generally result in a higher attack AUC-ROC.
> - Comparing the different curves in the right-most subfigure of Figure 5 (which represents various adaptation techniques from top to bottom), the curves for techniques with more trainable parameters are positioned higher, indicating greater privacy leakage.
>
> Both are consistent with our conclusion.
>
> **[2. It shows in Figure 3 that except for Likelihood-ref and Gradient Norm, other member inference attacks couldn’t achieve relatively high performance. It would be good to see some explanations about the difference.]**
>
> We greatly appreciate the suggestion and will enrich the discussion in the revision accordingly. We hypothesize that the primary advantage of certain attack methods may largely stem from their more privileged threat model, which assumes greater knowledge accessible to the adversary, as briefly discussed in Section 2.

---

### Decision · Program_Chairs · 2024-09-26

**Decision:**

Accept (Spotlight)

**Comment:**

This benchmark paper provides a systematic evaluation on the privacy leakage during adaptation of LLMs, through the lens of membership inference attacks.

This paper is strong in the following aspects: 1) comprehensive evaluation settings in terms of model architecture, adaption techniques, attack methods, and evaluation metrics; 2) clear statement of research questions and design of experiments that answer the questions; 3) valuable observations like that larger models tend to have higher privacy vulnerability.
Other than making the necessary improvements suggested by the reviewers, I would also suggest adding additional evaluations on other privacy attacks like data reconstruction and PII inference, than the MIA attacks.

Typo:  “Figure ??” in Section 5.5.